# The Anti-Obesity Effect of Porous Silica Is Dependent on Pore Nanostructure, Particle Size, and Surface Chemistry in an In Vitro Digestion Model

**DOI:** 10.3390/pharmaceutics14091813

**Published:** 2022-08-29

**Authors:** JingYi Chen, John P. Hanrahan, Joe McGrath, Melissa A. Courtney, Clive A. Prestidge, Paul Joyce

**Affiliations:** 1UniSA Clinical & Health Sciences, University of South Australia, Adelaide, SA 5000, Australia; 2Glantreo Limited, ERI Building Lee Road, T23 XE10 Cork, Ireland

**Keywords:** obesity, mesoporous silica, biomaterials, digestive enzymes, lipid digestion

## Abstract

The potential for porous silica to serve as an effective anti-obesity agent has received growing attention in recent years. However, neither the exact pharmacological mechanism nor the fundamental physicochemical properties of porous silica that drive its weight-lowering effect are well understood. Subsequently, in this study, an advanced in vitro digestion model capable of monitoring lipid and carbohydrate digestion was employed to elucidate the effect of porous silica supplementation on digestive enzyme activities. A suite of porous silica samples with contrasting physicochemical properties was investigated, where it was established that the inhibitory action of porous silica on digestive enzyme functionality was strongly dependent on porous nanostructure, particle size and morphology, and surface chemistry. Insights derived from this study validate the capacity of porous silica to impede the digestive processes mediated by pancreatic lipase and α-amylase within the gastrointestinal tract, while the subtle interplay between porous nanostructure and enzyme inhibition indicates that the anti-obesity effect can be optimized through strategic particle design.

## 1. Introduction

Obesity is a rapidly growing concern worldwide, with the World Health Organization (WHO) estimating that 1.9 billion adults are overweight or obese [1]. In most cases, obesity is a preventable disease driven by dietary imbalances between energy consumed and expended [2]. Despite the preventable nature of the disease, altering the public perception relating to dietary habits and the importance of physical exercise has proven challenging, with the prevalence of obesity continuing to rise at rapid rates. Since obesity is linked with a myriad of comorbidities that significantly decrease life expectancy, including Type II Diabetes, cardiovascular disease, and cancer, there exists an urgent need for new therapeutics that effectively treat obesity through safe mechanisms of action.

Orlistat remains one of the leading anti-obesity therapeutics on the market, and functions by inhibiting gastric and pancreatic lipase activity within the gastrointestinal (GI) tract [3]. In doing so, fat digestion is inhibited, restricting the capacity of fat to be absorbed across the intestinal epithelium into the systemic bloodstream. This mechanism of action leads to counterproductive and significant adverse effects, including diarrhea and GI distress, due to the process of undigested fats and lipids passing through the colon [4]. Ultimately, the adverse side effects linked with orlistat have led to varying patient efficacies in promoting weight loss, with several clinical studies reporting poor patient adherence due to the discomfort caused by taking orlistat before consuming a high-fat content meal [5,6]. 

The limitations linked with orlistat present opportunities for the development of new anti-obesity therapeutics that act locally within the GI tract to limit energy intake, but through a mechanism of action with limited side effects. In recent years, it has been hypothesized that porous colloid biomaterials (e.g., smectite clays [7,8], porous silica [9,10,11], activated carbon [12]) may serve as effective anti-obesity therapies through their capacity to adsorb large quantities of macronutrients within their porous matrices [13]. In contrast to orlistat’s mechanism of action where the inhibition of lipase activity leads to the production of oily stools, porous colloids can inhibit lipase activity while also adsorbing remaining undigested fats within the GI tract, thus eliminating a large portion of the diarrhea and GI distress caused by inhibiting GI lipolysis [8].

Porous silica is one such colloid that has received increasing attention for its anti-obesity potential, with oral dosing triggering reductions in metabolic risk factors in both animal obesity models [9,14,15] and human clinical studies [11,16]. Importantly, in a Phase I human clinical study, it was shown that oral dosing of porous silica did not trigger any abdominal discomfort, with minimal and inconsistent changes in bowel habits reported, compared to a placebo control [17], thus highlighting the safety and tolerability of this as an anti-obesity therapy, which is a fundamental consideration given the GI complications associated with orlistat. However, the exact mechanism of action of porous silica is still poorly understood, and little is known with respect to the key physicochemical properties that promote a reduction in energy intake and subsequent weight gain reductions [15]. 

Kupferschmidt et al. [10] were the first to report that the anti-obesity effect of porous silica is pore size-dependent, with SBA-15 mesoporous silica comprised of cylindrical pores with a mean pore width of 11 nm demonstrating the capacity to significantly reduce weight gain in mice fed a high-fat diet over a 12-week period. In contrast, NFM-1 mesoporous silica with a mean pore width of 2 nm was unable to prevent weight gain in the same animal model. It was hypothesized that this pore size effect was due to the capacity of mesoporous silica with sufficiently large pores to adsorb lipase and restrict its capacity to interact with the lipid-in-water interface, therefore inhibiting lipid digestion [10]. However, further studies revealed that pore size is not the only factor that is important for an anti-obesity effect, since a study by Rinde et al. [14] revealed varying efficacies for four different mesoporous silica, despite each silica type having comparable pore sizes of ~10 nm. From this, it was speculated that the pore structure, connectivity, and size distribution may all play important roles in promoting weight gain reductions.

Additional studies are required to deconvolute the impact of fundamental physicochemical properties on the capacity for porous silica to impede with digestive processes within the GI tract, to ultimately restrict lipid absorption and calorific intake. In this study, an in vitro digestion model that simulates the gastrointestinal environment following the consumption of a high macronutrient content meal was employed to systematically investigate the impact of key physicochemical properties on digestive enzyme activity. Focus was attributed to the following key parameters: porosity, pore width, particle shape and morphology, and surface chemistry. Thirteen silica samples were tested and both time-dependent lipid and carbohydrate digestion were monitored, allowing key insights to be derived relating to the optimal particle structure for inhibiting digestion. The findings from this study are fundamentally important for understanding the mechanism of action of porous silica in promoting an anti-obesity effect. This improved understanding is critical for the future development and clinical translation of oral porous silica therapies as a treatment approach for obesity and other metabolic disorders.

## 2. Materials and Methods

### 2.1. Materials

All porous silica samples used throughout this study were supplied by Glantreo Ltd. (Cork, Ireland), including: SOLAS™ spherical porous silica microparticles (PSM) with varying pore and particle sizes (labelled PSM-1 to PSM-5 throughout), SOLAD™ non-porous silica microparticles (NPSM) with varying particle sizes, SBA-15 mesoporous silica with varying pore sizes and surface chemistries (labelled SBA-15(1) to SBA-15(4) throughout), and MCM-41 with varying surface chemistries (labelled MCM-41(1) to MCM-41(2) throughout). Full details of the silica samples used in this study are provided in Table 1. Medium-chain triglycerides (MCT; Miglyol 812^®^) were obtained from Hamilton Laboratories (Adelaide, Australia). FaSSIF/FeSSIF/FaSSGF powder for lipolysis studies was purchased from Biorelevant.com Ltd. (London, UK). Lipase from *Candida antarctica* (6000 TBU/mL), starch (soluble), cellulose, sodium hydroxide pellets, glacial acetic acid, sodium chloride, 4-bromophenylboronic acid (4-BBA), deuterated chloroform (CDCl_3_), and phosphate buffered saline (PBS) tablets were supplied by Sigma-Aldrich (Castle-Hill, Australia). Porcine pancreatin extract (activity equivalent to 8 × USP specification) was supplied by MP Biomedicals (Seven Hills, Australia). All chemicals and solvents were of analytical grade and used as received. High-purity Milli-Q water was used throughout the study.

### 2.2. Physicochemical Characterization of Porous Silica Materials

#### 2.2.1. Nitrogen Adsorption/Desorption Isotherms

Nitrogen isotherms were measured at liquid nitrogen temperature (−196 °C) using a Micromeritics TriStar II 3020 V1.01 volumetric adsorption analyzer (Micromeritics Instrument Corporation, GA, USA). Prior to measuring, the MPS particles were outgassed for 3 h at 200 °C. The Brunauer-Emmett-Teller equation was used to calculate the surface area from the adsorption data obtained in the relative pressure range of 0.05 to 0.3 [18]. The total pore volume was calculated from the amount of gas adsorbed at 0.91 (*P/P*_0_), and the pore size distribution curves were derived using Barrett-Joyner-Halenda (BJH) analysis.

#### 2.2.2. Scanning Electron Microscopy (SEM)

The particle size and surface morphology of PSM, NPSM, SBA-15, and MCM-41 samples were studied by high-resolution analytical SEM (Zeiss Merlin, Oberkochen, Germany). Samples were mounted on double-faced carbon tape and sputter coated with ~10–20 nm platinum prior to imaging at an accelerating voltage of 1–2 kV.

#### 2.2.3. Particle Sizing

The average particle size of each silica sample (refractive index = 1.52) was assessed after redispersion in intestinal buffer (refractive index = 1.33) using laser diffraction analysis (Malvern Mastersizer, Worcestershire, UK).

### 2.3. Gastrointestinal Digestion Studies Using an In Vitro Obesity Model

#### 2.3.1. Preparation of Simulated Gastric and Intestinal Digestion Buffers

A two-step in vitro digestion model that simulates the gastric and intestinal environment following consumption of a high-fat, high-carbohydrate content meal was used to investigate the impact of silica physicochemical properties on lipid and starch digestion [9]. Fed-state simulated gastric fluid (FeSSGF) was prepared with biorelevant concentrations of bile salts (0.08 mM) and phospholipids (0.02 mM) by dissolving FaSSIF/FeSSIF/FaSSGF powder (0.06 g/L) and sodium chloride (2.00 g/L) in Milli-Q water. To simulate the early to intermediate phases of gastric lipolysis [19], the pH of FeSSGF was adjusted to 5.00 ± 0.01 with sodium hydroxide (1 M) and/or hydrochloric acid (1 M) and the mixture was stirred for ~4 h to allow for complete dissolution of buffer excipients. 

Fed-state simulated intestinal fluid (FeSSIF) was prepared with biorelevant concentrations of bile salts (15 mM) and phospholipids (3.75 mM) by dissolving FaSSIF/FeSSIF/FaSSGF powder (11.2 g/L) in intestinal buffer containing sodium hydroxide (4.04 g/L), glacial acetic acid (8.65 g/L) and sodium chloride (11.9 g/L). The pH of FeSSIF was adjusted to 6.00 ± 0.01 with sodium hydroxide (1 M) and/or hydrochloric acid (1 M), and the mixture was stirred for ~ 4 h to allow for complete dissolution and stabilization of buffer excipients. Note: pH 6.00 ± 0.01 was selected for FeSSIF in this study to maintain adequate pancreatin activity and to ensure monitoring of fatty acid release was possible.

Pancreatin extracts were prepared by stirring 2 g of porcine pancreatin powder in 10 mL intestinal buffer for 15 min, followed by centrifugation at 2268× *g* and 4 °C for 20 min. The supernatant phase was collected for use within GI lipolysis studies.

#### 2.3.2. Quantifying Lipid Digestion under Simulated Fed Conditions

A two-step in vitro GI lipolysis model was employed under simulated fed-state conditions to mimic the GI environment during digestion of a high-fat, high-carbohydrate content meal. Firstly, MCT (625 mg), starch (625 mg), and cellulose (31 mg) were dispersed within FeSSGF (10 mL) through continuous stirring at 600 rpm for 10 min in a thermostated glass reaction vessel (37 °C). To this dispersion, silica samples were added at 10% *w*/*w* relative to the lipid content (i.e., 62.5 mg). The pH of the lipolysis medium was readjusted with 0.1 M NaOH or HCl to 5.00 ± 0.01, prior to initiation of simulated gastric lipolysis via addition of lipase from *Candida antarctica* (100 µL; equivalent to 600 TBU). Gastric lipolysis was monitored for 30 min, where the pH was continuously adjusted with 0.6 M NaOH using a pH-stat titration unit (902 Titrando, Metrohm, Switzerland) to maintain a constant pH within the digestion media. Note: gastric lipolysis data is not reported. Upon completion of gastric lipolysis, FeSSIF (20 mL) was added and the pH was allowed to slowly adjust to 6.00 ± 0.01 via auto-titration over a 10 min period. Intestinal lipolysis was initiated through the addition of 2 mL pancreatin extract (containing ~2000 TBU of pancreatic lipase activity and ~1600 TBU of α-amylase activity), prepared within intestinal buffer. Intestinal lipolysis was monitored for 60 min, and the rate and extent of free fatty acid production was calculated with respect to the amount of NaOH titrated to maintain a constant pH of 6.00 ± 0.01 during intestinal lipolysis. Lipase activity was calculated using the following equation:(1)$$Lipase\;activity=\frac{AULC_{PS}}{AULC_{No\;silica}}\times 100$$
where *AULC_PS_* is the area-under-the-lipolysis curve (*AULC*) in the presence of a porous silica sample, and *AULC_No silica_* is the *AULC* in the absence of any porous silica sample.

#### 2.3.3. Quantifying Starch Digestion under Simulated Fed Conditions

Aliquots (500 µL) were periodically collected in Eppendorf tubes throughout the in vitro digestion experiment and were immediately immersed within a water bath at 90 °C for 10 min for inhibition of amylase activity. The concentration of starch digestion products, specifically glucose and maltose, was analyzed using a HPLC system (Shimadzu Corporation, Kyoto, Japan) consisting of a series of LC-20ADXR pumps, SIL-20ACXR auto sampler, CTO-20AC column oven set at 30 °C, an ELSD-LTII evaporative light scattering detector, and a Luna Omega Sugar 100 Å analytical column (3 µm, 4.6 mm ID x 250 mm; Phenomenex). The mobile phase was a mixture of acetonitrile and Milli-Q water (80:20 *v*/*v*) eluted at a flow rate of 1.0 mL/min. The limit of detection (LOD) of the analytical method was 20 µg/mL for each sugar. Linear calibration curves (R^2^ ≥ 0.99) were plotted for chromatographic peak areas against glucose and maltose concentrations over the range of 33–1000 µg/mL, without the additional of an internal standard. Digestion aliquots were diluted suitably to meet the calibration curve. The effective glucose concentration was determined by factoring in that maltose is comprised of two glucose monomers, thus the quantified glucose and maltose mass concentrations were combined as a factor of glucose molar mass. Amylase activity was calculated using the following equation:(2)$$Amylase\;activity=\frac{AUSC_{PS}}{AUSC_{No\;silica}}\times 100$$
where *AUSC_PS_* is the area-under-the-starch digestion curve (*AUSC*) in the presence of a porous silica sample, and *AUSC_No silica_* is the *AUSC* in the absence of any porous silica sample.

#### 2.3.4. Calculating Enzyme Inhibitory Response

The degree of overall enzyme inhibition (Inhibitory Response; IR) was calculated as a function of lipase and amylase activities, using Equation (3) below, to give a predictor of the porous silica samples that triggered the greatest in vitro anti-obesity effect. Calculating IR relies on the assumption that the inhibition of both lipase and amylase is equally important for triggering an anti-obesity effect, where IR ≤ 0 is equal to no enzyme inhibition and IR = 1 is equal to the maximum enzyme inhibition.
(3a)$$Inhibitory\;Response=\frac{1}{2}\times \left(\frac{AULC_{No\;silica}-AULC_{PS}}{AULC_{No\;silica}}+\frac{AUSC_{No\;silica}-AUSC_{PS}}{AUSC_{No\;silica}}\right)$$
(3b)$$Inhibitory\;Response=\frac{1}{2}\times \left(\frac{100-Lipase\;activity}{100}+\frac{100-Amylase\;activity}{100}\right)$$

### 2.4. Quantifying Organic Media Adsorption Using Thermogravimetric Analysis (TGA)

The organic content adsorbed by porous silica samples following in vitro digestion studies was determined using TGA (Discovery TGA, TA Instruments, New Castle, DE, USA). Upon digestion completion, 5 mL of media was withdrawn and inhibited with 50 μL 4-BBA (0.5 M in methanol), prior to centrifugal separation at 22,000× *g* for 5 min. The obtained pellets were dried under vacuum at 40 °C for 72 h prior to analysis. TGA thermograms were acquired by heating samples at a scanning rate of 10 °C/min from 20 to 600 °C under a nitrogen gas atmosphere. The lipid and carbohydrate content decomposed in the temperature range of 150–450 °C. The weight loss, after correction for water content, was computed using the associated TA Universal Analysis software, which corresponded to organic material adsorbed within each porous silica sample. The silica content remained entirely stable within this temperature range. It should be noted that other insoluble, organic media within the lipolysis media (e.g., bile salts, enzymes) may also decompose within an equivalent temperature range to lipid and carbohydrates. However, it was assumed that the majority of organic species adsorbed by the porous particles related to MCT, starch, and cellulose.

### 2.5. Statistical Analysis

The data obtained within this study were tested for statistical significance using an unpaired *t*-test, one-way ANOVA with Dunnett’s multiple comparisons tests, and two-way repeated measurements ANOVA, followed by Tukey’s multiple comparisons test (GraphPad Prism version 7.03, GraphPad Software Inc., San Diego, CA, USA). The level of significance was set at *p* < 0.05.

## 3. Results & Discussion

### 3.1. Physicochemical Characterization of Porous Silica Particles

A suite of porous silica samples with varying porosities, pore widths and geometries, particle sizes, and surface chemistries were compiled for this study, with the key physicochemical properties presented in Table 1. Each silica sample has been categorized according to its primary physicochemical properties; these samples are porous silica microparticles (PSM), non-porous silica microparticles (NPSM), Santa-Barbara Amorphous-15 (SBA-15), and Mobil Composition of Matter-41 (MCM-41). As can be observed through scanning electron micrographs in Figure 1, the surface morphology of each of these particle types differs significantly, with PSM and NPSM being comprised of spherical microparticles of varying particle sizes between ~2 µm and ~13 µm, in contrast to the random rod-shaped structure of SBA-15 and agglomerate structure of MCM-41.

### 3.2. In Vitro Digestion Studies

#### 3.2.1. The Impact of PSM Pore Size on Digestive Enzyme Activity

In vitro digestion studies were performed under GI conditions that simulated the fed state following consumption of a high-fat, high-carbohydrate meal to elucidate the impact of silica physicochemical properties on digestive enzyme activity (specifically, lipase and amylase activity). A two-step GI model where gastric digestion was modelled for 30 min, followed by a 60 min intestinal phase, was employed based on previous studies [20,21]. This model ensures that the particles and digestion media are exposed to biologically relevant gastric and intestinal media; however, only intestinal digestion data is presented here due to minimal digestion occurring within the gastric phase [9]. Lipid digestion was monitored through titration of fatty acids, produced through the hydrolysis of glycerides, with sodium hydroxide using a pH-stat autotitrator, while starch digestion was monitored by quantifying the glucose and mannose concentration within digestion media, which are produced through amylase-mediated digestion of starch. A schematic overview of the experimental approach is provided in Figure 2.

In all studies, the silica samples were dispersed within the gastric buffer at 10 wt% relative to both the lipid and carbohydrate content. PSM-1 to PSM-4 were selected as ideal silica samples for assessing the impact of pore width on digestive enzyme activity, given their spherical and uniform morphologies and narrow particle size distributions. For all samples, including the control (i.e., no silica added), time-dependent lipolysis profiles revealed biphasic digestion kinetics where fatty acid release did not plateau within the 60 min digestion period (Figure 3). In the absence of silica, the final extent of fatty acid release was 14.7 ± 1.5 mM with an AUC of 586 ± 32 mM/min, which represents 100% lipase activity according to Equation (1). Lipase activity was clearly impacted by the addition of PSM to digestion media, in a pore width-dependent manner, whereas PSM-1 (pore width = 2.30 nm) and PSM-4 (pore width = 23.6 nm) did not impact lipase activity. In contrast, PSM-2 (pore width = 6.71 nm) and PSM-3 (pore width = 10.2 nm) significantly impacted lipase activity through reductions to 66.0 ± 10% and 79.4 ± 4.2%, respectively (Figure 3B). It has been previously hypothesized that an optimal pore size exists for inhibiting lipid digestion, based on the ability of porous silica to adsorb lipase in a restricted confirmation where it is unable to access the lipid-in-water interface [10]. The current findings support this hypothesis, since the pore size distributions of PSM-1 and PSM-4 appear too small and too large, respectively, to interfere with lipase action. Since lipase has a molecular diameter of approximately 4–5 nm, the pore widths of PSM-2 and PSM-3 are more optimal for adsorbing lipase and restricting its capacity to interact with the lipid-water interface. Thus, these findings indicate that at least one mechanism of inhibitory action for porous silica on lipase activity is through the adsorption of the enzyme within the porous matrix.

The rate and extent of amylase-mediated starch digestion were also impacted by PSM in a pore width-dependent manner, where PSM-3 had the optimal pore size for inhibiting starch digestion, reducing amylase activity to 71.6 ± 2.2% (Figure 3). This suggests that the optimal pore size for inhibiting amylase activity is greater than that for lipase inhibition, which is in accordance with the relative molecular dimensions of each enzyme. Amylase is a larger enzyme, with egg-shaped dimensions of approximately 7 nm long by 4 nm wide. Thus, the pore size required to entrap and inhibit amylase is greater than that required for lipase. A previous study by Waara et al. [11] revealed that amylase activity was not inhibited by porous silica with pore size distributions below 7 nm. However, that was not the case in this study, with PSM-1 and PSM-2 significantly impacting amylase activity despite having pore sizes of 2.30 and 6.71 nm, respectively. This suggests that the inhibitory mechanism of porous silica may not be solely due to the entrapment of digestive enzymes within the porous matrices, but through multiple alternate mechanisms where the presence of solid colloids within the digestion media physically impedes the capacity of digestive enzymes to access the substrate [22]. However, this interplay between various mechanisms is likely subtle, since PSM-4 was not shown to inhibit amylase activity (rather activity was enhanced), which suggests that amylase was free to diffuse into and out of the pores. Furthermore, the available surface area of PSM-4 was significantly less than that of PSM-1, 2, and 3, which may restrict its capacity to interfere with digestive enzyme activity through various mechanisms.

#### 3.2.2. The Impact of PSM Pore Size on Organic Media Adsorption

An alternate mechanism by which porous silica can induce anti-obesity effects is through its capacity to adsorb large quantities of digested and undigested macronutrients from the GI tract, thus preventing their absorption into the systemic bloodstream [9]. Interestingly, previous studies have indicated that the adsorption of lipid-based organic media by porous colloids correlates more strongly with a reduction in weight gain, compared to correlations between lipase activity and weight gain [9]. Subsequently, the capacity for PSM to adsorb lipid and carbohydrates during in vitro digestion was investigated by quantifying the mass of organic matter adsorbed relative to the mass of inorganic matter (i.e., silica), using TGA. At the completion of the digestion period, PSM-1 adsorbed the greatest degree of organic matter, with over 150 mg adsorbed, which corresponds to >2.5-fold its own weight. A pore width- and surface area-dependent effect was observed (Figure 4), where particles with a smaller pore width and greater surface area promoted adsorption of organic media. These findings indicate that PSM with small pore size distributions are capable of adsorbing high degrees of organic matter related to energy intake, despite their inability to impact digestive enzyme functionality. 

#### 3.2.3. The Impact of SBA-15 Pore Size on Digestive Enzyme Activity

SBA-15 mesoporous silica has been the most widely investigated porous silica type for use as an anti-obesity therapeutic [9,10,11,15]. Unlike PSM, which are comprised of uniform microparticles (~2 µm in diameter) with random pore arrangements, SBA-15 forms large fiber-like aggregates with organized hexagonal pore structures that extend along the length of the individual rod-shaped particles (Figure 1). This unique porous nanostructure of SBA-15 presents a longer diffusional path length for digestion media (including enzymes, bile salts, lipids, etc.,) to be entrapped [11,23,24]. Here, it was evident that the rate and extent of fed state lipolysis was inhibited by each SBA-15 sample, when compared to the control, where lipase activity was reduced to 65–85% depending on the sample (Figure 5A,B). The impact of pore size was not statistically significant for SBA-15; however, the current findings suggest that increasing the pore size to ~9 nm does promote a greater inhibitory response. However, previous studies have indicated that lipase activity is enhanced in SBA-15 as a function of increasing microporosity [24]. Since SBA-15(3) has almost no micropores (i.e., micropore area = 1.45 m^2^/g), this may serve as a contributing factor to the decreased lipase activity in the presence of SBA-3 (rather than pore size alone), when compared to SBA-15(1) and SBA-15(2) which have micropore areas >150 m^2^/g. When comparing lipolysis profiles of PSM-3 and SBA-15(3) (i.e., silica samples with comparable pore sizes, but varying nanostructures), it is evident that SBA-15 does inhibit lipase activity to a greater degree than PSM, as shown through lipase activities of 79.4 ± 4.2% and 67.4 ± 13.2%, respectively. This suggests that the increased diffusional path length of SBA-15 may entrap lipolysis media to a greater degree than PSM [24].

As was the case for lipid digestion, starch digestion was also inhibited when supplemented with each SBA-15 sample (Figure 5C,D); however, the inhibitory response triggered by SBA-15(1) with a mean pore width of 3.55 nm was minimal, suggesting that the pores were too small for amylase entrapment. Despite this, a reduction in amylase activity to 93.8 ± 2.9% by SBA-15(1) further confirms that inhibition is not driven solely by enzyme entrapment. In contrast to SBA-15(1), amylase-mediated starch hydrolysis was significantly impeded in the presence of SBA-15(2) and SBA-15(3) with pore sizes of 5.99 and 8.74 nm, respectively. This finding supports previous studies that have highlighted a pore size-dependent effect in amylase entrapment and inhibition, where SBA-15 with sufficiently large enough pore sizes for amylase to freely diffuse into promote a greater reduction in starch digestion [11]. It is of interest that SBA-15(2) with a mean pore width of 5.99 nm led to a statistically equivalent reduction in amylase activity compared to SBA-15(3) with a mean pore size of 8.74 nm, since the mean pore width of SBA-15(2) is less than the largest molecular dimension of amylase (i.e., ~7 nm). This suggests that amylase may be capable of conformational changes or can still freely diffuse longitudinally throughout pores that confine its molecular rotation.

#### 3.2.4. The Impact of Particle Size on In Vitro Lipid Digestion

The impact of particle size on in vitro lipolysis was systematically investigated in this study by exposing digestion media to both non-porous and porous silica microparticles with comparable physicochemical properties (e.g., pore size), but contrasting particle sizes. The impact of particle size on starch digestion was also investigated, but no effect was observed, and as a result, data is not presented here. A decrease in particle size was shown to positively correlate with a reduction in lipase activity for both NPSM and PSM (Figure 6). Interestingly, NPSM-2 with a mean particle size of 560 nm promoted the greatest reduction in lipase activity (65.1 ± 4.7%) out of all the silica samples tested within this study. This highlights the multifaceted mechanism of lipase inhibition and further emphasizes that the proposed anti-obesity effect of silica is not solely dependent on entrapment of enzymes within their porous matrices. Previous studies have highlighted the capacity for colloids to interfere with lipase action by adsorbing to the lipid-in-water interface [24,25]. Since lipase is an interfacially active enzyme, whereby it is activated through adsorption onto an interface, this retardation of bioaccessibility for lipase to the lipid-in-water interface inhibits digestion of triglycerides into fatty acids [26]. Thus, smaller particles are more capable of tightly packing onto the lipid-in-water interface, physically shielding enzyme access. This phenomenon is emphasized here and is shown to be independent of particle porosity, and hence, particle size should be considered a significant contributor to the anti-obesity effect of silica.

#### 3.2.5. The Impact of Surface Modifications on In Vitro Lipid Digestion

The impact of surface chemistry on in vitro lipid digestion was investigated by comparing unmodified SBA-15 and MCM-41 mesoporous silicas with dehydroxylated SBA-15 and aluminum-doped MCM-41, respectively. In both instances, unmodified silica samples promoted a greater reduction in lipase activity compared to the surface-modified counterparts (Figure 7A,B). This is considered to be due to the enhanced capacity for surface-modified SBA-15 and MCM-41 to adsorb organic content from the digestion media, as highlighted in Figure 7C,D. That is, lipid digestion products, being free fatty acids and monoglycerides, are highly surface-active, amphiphilic molecules that adsorb to the lipid-in-water interface during lipolysis [27]. These digestion products subsequently impede lipase activity as a self-limiting interaction [28]. However, the presence of porous colloids within the digestion media that remove fatty acids from the lipid-in-water interface through adsorptive processes can promote lipase bioaccessibility of the interface, which ultimately promotes enhanced lipolysis kinetics [20]. The adsorption of organic media at the completion of lipid digestion studies was enhanced for both dehydroxylated SBA-15 and Al-doped MCM-41 compared to their unmodified alternatives (Figure 7C,D). This is likely due to the increased hydrophobicity for dehydroxylated SBA-15 and increased cationic charge of Al-doped MCM-41, which is expected to increase fatty acid adsorption through both hydrophobic and electrostatic interactions, respectively. The increased adsorption of lipid digestion products onto surface-modified silica is expected to promote lipase activity, but may serve as an alternate mechanism for promoting an anti-obesity effect, despite their inability to limit lipase activity. Further studies, specifically through the use of in vivo anti-obesity models, are required to validate this and compare the anti-obesity effect of unmodified and surface-modified porous silica.

### 3.3. Enzyme Inhibitory Response to Silica Samples

The enzyme Inhibitory Response (IR) to each silica sample was calculated by factoring the combined impact of silica exposure on lipase and amylase activities, in an attempt to predict the silica samples that provided the greatest in vitro anti-obesity response. N.B., this calculation relies on the assumption that the inhibitions of both lipase and amylase contribute equally to an anti-obesity effect. Further in vivo investigations are required to validate this approach. Table 2 highlights the lipase and amylase activities and the corresponding IR for each silica sample. It can be observed that the greatest IR was triggered by PSM-2 and SBA-15(3), with over a quarter of enzyme activity being inhibited in the presence of these two porous silica samples (i.e., IR > 0.25). Further, IR values highlight the importance of the physicochemical properties of silica on triggering an in vitro anti-obesity effect, with the greatest IR being observed for porous silica samples with a pore size between 5.99 nm and 10.2 nm. These findings suggest that these particles are optimal for further detailed investigations to pre-clinically validate their anti-obesity potential.

### 3.4. Opportunities, Limitations, and Future Directions

The current study highlights the unique and complex interplay between the physicochemical properties of silica and their potential anti-obesity effects via manipulation of GI digestive processes. While previous studies have validated this anti-obesity effect in vivo, few fundamental in vitro studies have been performed to elucidate the mechanism/s of action of porous silica in promoting a reduction in metabolic risk factors. This study aimed to address this knowledge gap through employing an advanced in vitro digestion model that simulated the GI environment following consumption of a high-fat, high-carbohydrate content meal. However, this approach introduces limitations, since mimicking and simulating the complex physiological processes related to digestion through an in vitro model remains challenging. The limitations of in vitro digestion models have been discussed previously [9,29], one of which is the oversimplification of monitoring solely lipid digestion as a predictor of energy intake. In an attempt to circumvent this limitation, carbohydrate digestion and the degree of organic matter adsorption was also monitored in the current study, but this still potentially ignores other mechanisms by which porous silica may induce an anti-obesity effect (e.g., microbiome interactions, anti-inflammatory mechanisms).

In light of such limitations, future studies are required to further enhance the understanding related to the anti-obesity effect of porous silica. In particular, we propose performing long-term obesity studies using an in vivo model of obesity where rodents are fed a high-fat, high-sugar diet that more accurately simulates the Western diet (rather than only a high-fat diet) [30,31]. Within these future studies, it is critical that a suite of porous silica samples with varying physicochemical properties (e.g., porous nanostructure, particle size and morphology, surface chemistry) are investigated to determine if correlations exist between the in vitro findings in the current study and in vivo pharmacodynamics. Previous focus has been solely attributed to SBA-15 mesoporous silica [10,11], but the current study highlights the capacity for alternate porous silica samples to also induce an anti-obesity effect through inhibition of digestive enzyme function. Further, broad-spanning pharmacodynamics related to metabolic risk factors, beyond weight gain, should also be monitored to provide greater insight into the anti-obesity mechanism of action of porous silica. By advancing the fundamental knowledge related to this application of porous silica, it can be expected that the optimal properties for inducing an anti-obesity effect will be identified, which will promote the rapid translation of porous silica anti-obesity therapies to the clinic.

## 4. Conclusions

Systematic investigations of the physicochemical properties of silica that impact in vitro lipid and starch digestion have revealed that pore size, particle size and morphology, and surface chemistry all play important roles in mediating enzyme activities when silica is supplemented to digestion media. The current findings indicate that the mechanism of action of silica in impeding enzyme activity is complex and multifaceted, and not solely dependent on the entrapment of enzymes within their porous matrices. However, it is clearly apparent that porous silica particles with pore widths between 6–10 nm are optimal for triggering an inhibitory response of both lipase and amylase activity. Future work is required to validate these findings in in vivo animal models of obesity to ensure that silica can be designed with optimal anti-obesity activities, so that this new and upcoming therapy can be translated to the clinic for the effective mitigation of metabolic risk factors.

## Figures and Tables

**Figure 1 pharmaceutics-14-01813-f001:**
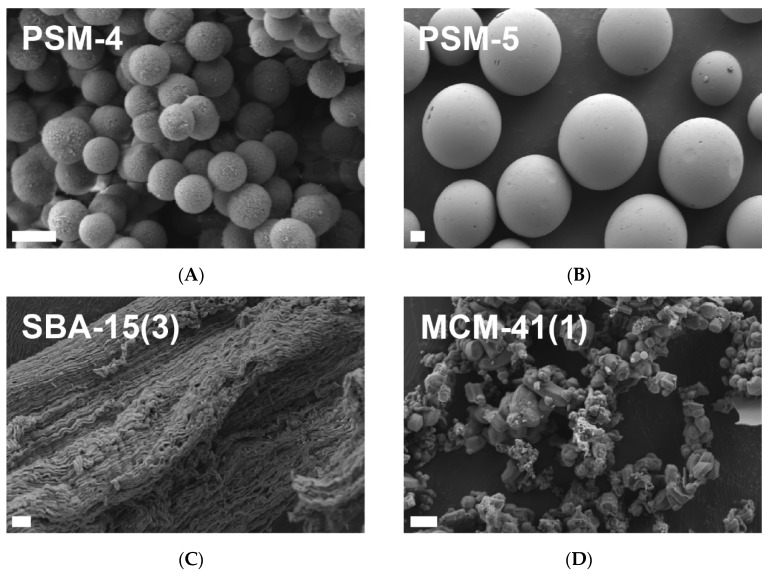
Scanning electron micrographs of (**A**) PSM-4 (SOLAS™ spherical porous silica particles with a mean pore width of 23.6 nm and mean particle size of 1.93 µm), (**B**) PSM-5 SOLAS™ spherical porous silica particles with a mean pore width of 10.1 nm and mean particle size of 12.6 µm), (**C**) SBA-15(3) (SBA-15 mesoporous silica with a mean pore width of 8.7 nm), and (**D**) MCM-41(1) (MCM-41 mesoporous silica with a mean pore width of 2.5 nm). Scale bars represent 2 µm.

**Figure 2 pharmaceutics-14-01813-f002:**
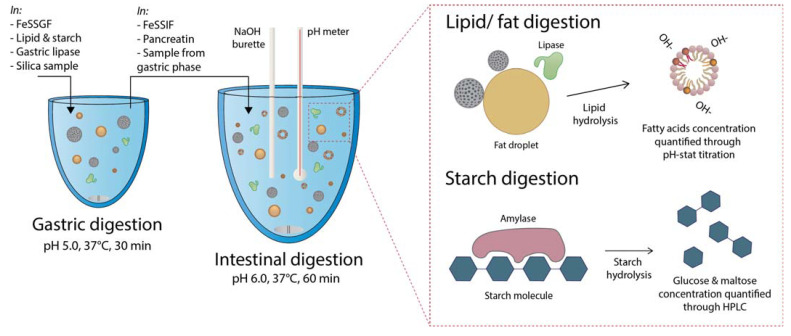
Schematic overview of the in vitro digestion approach.

**Figure 3 pharmaceutics-14-01813-f003:**
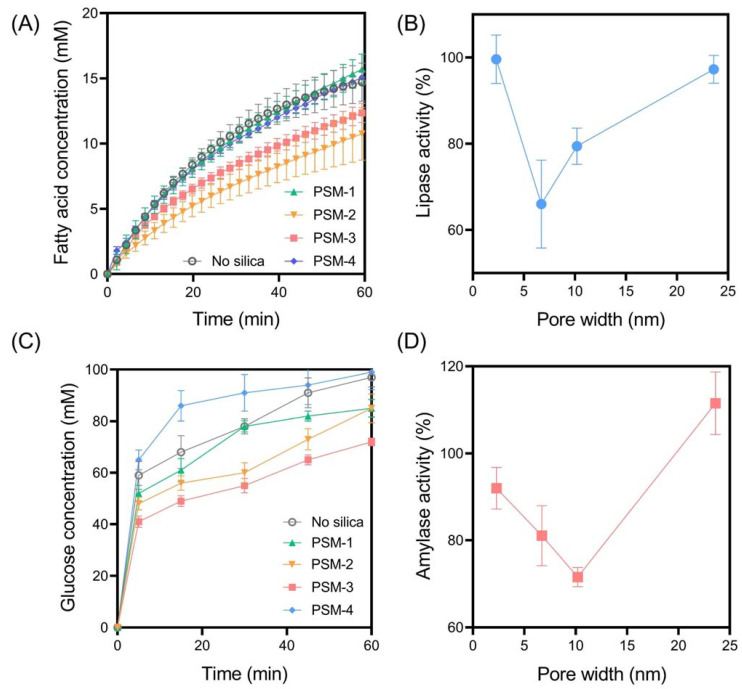
(**A**) In vitro lipid digestion profiles for spherical porous silica microparticles (PSM) with comparable particle sizes but contrasting pore size distributions. (**B**) The impact of PSM pore size on lipase activity. (**C**) In vitro starch digestion profiles for various PSM and (**D**) the corresponding impact of PSM pore size on amylase activity. In vitro digestion studies were performed under simulated fed state conditions, with data representing the digestion occuring during the intestinal phase (pH 6.0). Data represent mean ± S.D. (*n* = 3).

**Figure 4 pharmaceutics-14-01813-f004:**
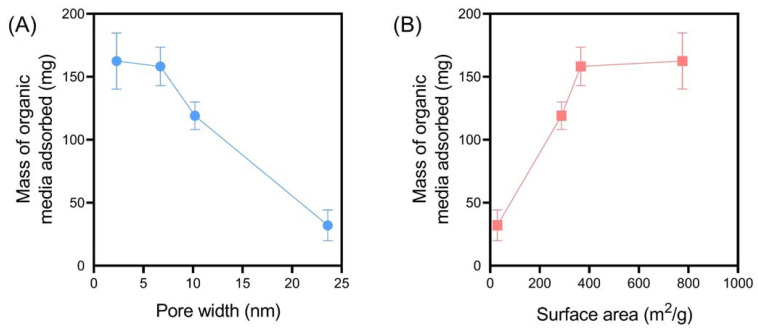
The impact of silica (**A**) pore size and (**B**) surface area on the extent of organic media adsorption at the completion of in vitro digestion studies for PSM, quantified using TGA. Data represent mean ± S.D. (*n* = 3).

**Figure 5 pharmaceutics-14-01813-f005:**
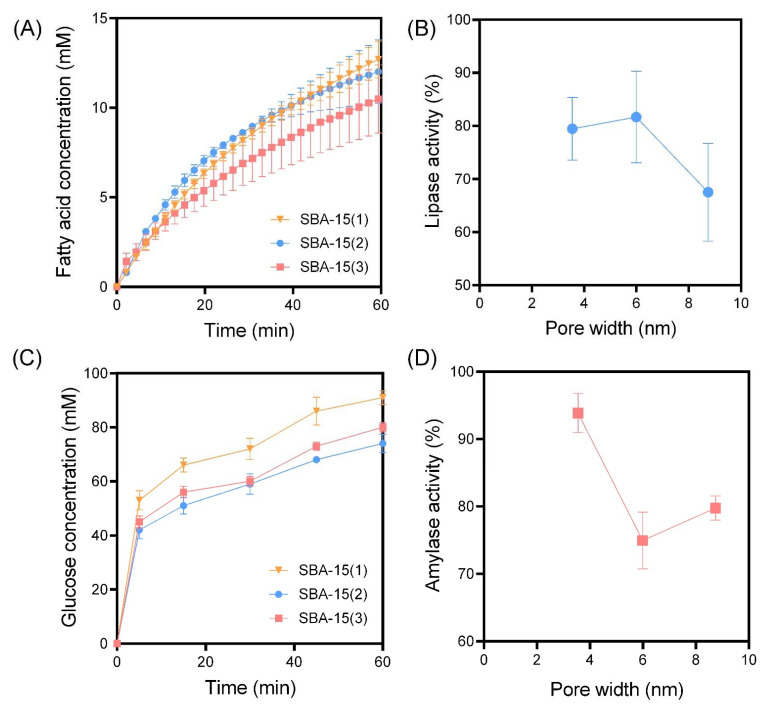
(**A**) In vitro lipid digestion profiles for SBA-15 mesoporous silica with contrasting pore size distributions. (**B**) The impact of SBA-15 pore size on lipase activity. (**C**) In vitro starch digestion profiles for various SBA-15 samples and (**D**) the corresponding impact of SBA-15 pore size on amylase activity. In vitro digestion studies were performed under simulated fed state conditions, with data representing the digestion occuring during the intestinal phase (pH 6.0). Data represent mean ± S.D. (*n* = 3).

**Figure 6 pharmaceutics-14-01813-f006:**
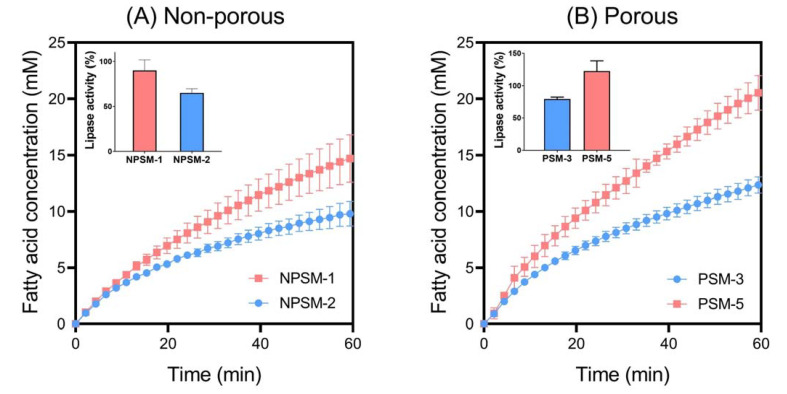
In vitro lipid digestion profiles highlighting the impact of particle size on lipolysis kinetics for (**A**) non-porous silica microparticles and (**B**) porous silica microparticles. Insets: lipase activity as a function of particle size. In vitro digestion studies were performed under simulated fed state conditions, with data representing the digestion occuring during the intestinal phase (pH 6.0). Data represent mean ± S.D. (*n* = 3).

**Figure 7 pharmaceutics-14-01813-f007:**
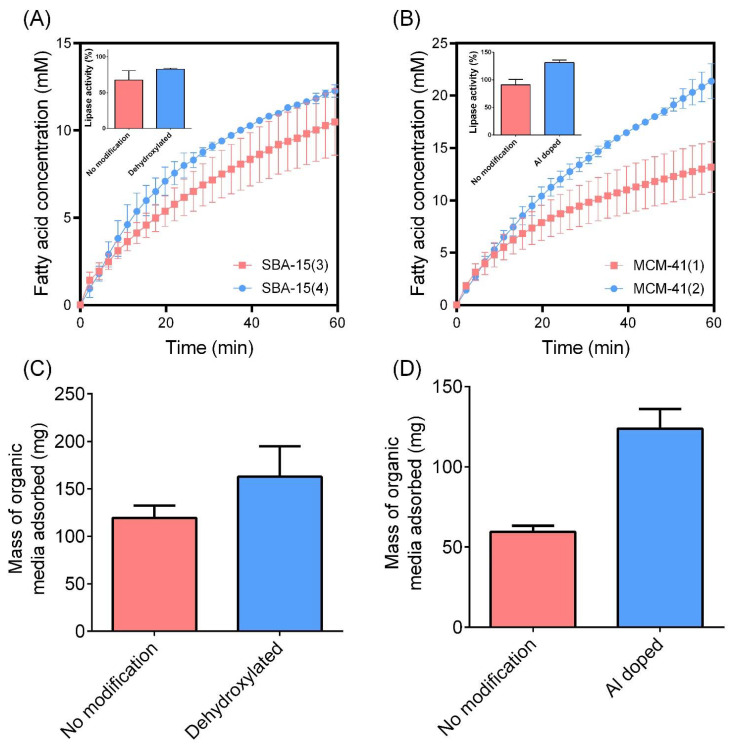
In vitro lipid digestion profiles highlighting the impact of surface chemistry on lipolysis kinetics for (**A**) SBA-15 and (**B**) MCM-41 mesoporous silica. Insets: lipase activity as a function of surface chemistry. The extent of organic media adsorption at the completion of in vitro digestion studies for (**C**) SBA-15 and (**D**) MCM-41 samples, quantified using TGA. In vitro digestion studies were performed under simulated fed state conditions, with data representing the digestion occuring during the intestinal phase (pH 6.0). Data represent mean ± S.D. (*n* = 3).

**Table 1 pharmaceutics-14-01813-t001:** Physicochemical properties of silica samples used in this study.

Silica Sample	Silica Type	Particle Shape	Surface Modification	Mean Pore width * (nm)	Total Surface Area ** (m^2^/g)	Micropore Area *** (m^2^/g)	Pore Volume (cm^3^/g)	Particle Size (µm)
PSM-1	Porous	Spherical	None	2.30	776	105	0.07	2.16
PSM-2	Porous	Spherical	None	6.71	366	61.2	0.63	2.03
PSM-3	Porous	Spherical	None	10.2	288	15.8	0.75	1.98
PSM-4	Porous	Spherical	None	23.6	29.2	1.27	0.61	1.93
PSM-5	Porous	Spherical	None	10.1	222	11.7	0.55	12.6
NPSM-1	Non-porous	Spherical	None	-	1.70	-	-	2.24
NPSM-2	Non-porous	Spherical	None	-	1.60	-	-	0.56
SBA-15(1)	Porous	Irregular	None	3.55	451	196	0.28	23.6
SBA-15(2)	Porous	Irregular	None	5.99	689	158	0.69	18.4
SBA-15(3)	Porous	Irregular	None	8.74	496	1.45	1.11	22.2
SBA-15(4)	Porous	Irregular	Dehydroxylated	8.74	496	1.45	1.11	19.9
MCM-41(1)	Porous	Irregular	None	2.47	1070	-	0.60	8.67
MCM-41(2)	Porous	Irregular	Al-doped	3.72	934	-	0.96	14.5

Note: * Mean pore widths are quoted from BJH distributions; ** Total surface area was calculated using the BET equation; *** Micropore area values were quoted from t-plot analysis at the thickness range from 3 to 5 Å.

**Table 2 pharmaceutics-14-01813-t002:** Enzyme activities and the corresponding Inhibitory Response (IR) when exposed to porous silica samples. Porous silica samples have been ordered from greatest to least enzyme inhibitory responses.

Silica Sample	Lipase Activity (%)	Amylase Activity (%)	Inhibitory Response, IR
PSM-2	66.0 ± 16	81.1 ± 6.9	0.265
SBA-15(3)	67.5 ± 13	79.8 ± 1.8	0.264
PSM-3	79.4 ± 2.9	71.6 ± 2.2	0.245
SBA-2	81.7 ± 12	74.9 ± 4.2	0.217
SBA-15(4)	82.7 ± 1.3	82.5 ± 6.7	0.174
NPSM-2	65.1 ± 4.7	101.8 ± 3.4	0.165
SBA-15(1)	79.5 ± 5.9	93.9 ± 2.9	0.133
PSM-1	99.6 ± 5.6	92.0 ± 4.8	0.042
NPSM-1	90.1 ± 12	102 ± 4.2	0.038
PSM-4	97.3 ± 3.2	112 ± 7.2	−0.044
PSM-5	123 ± 15	98.6 ± 3.8	−0.107

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
