# Peer review of "The Anti-Obesity Effect of Porous Silica Is Dependent on Pore Nanostructure, Particle Size, and Surface Chemistry in an In Vitro Digestion Model"

_pharmaceutics, 2022, doi:10.3390/pharmaceutics14091813_

Round 1

Reviewer 1 Report

The title of the paper is not suitable. The work concerns the influence of silica presence on the process of enzymatic degradation of lipids and starch. The title should be changed.

The designation of silicas as SBA-1, SBA-2, SBA-3 (SBA-15 type), MCM-1 (MCM-4a type) etc. is not correct and may be confusing for the Reader. As an example, the SBA-3 silica is well known and described in the literature as a distinct structure. The Authors should use other designation of samples, for example SBA-15(1), SBA-15(2) …., MCM-41(1), MCM-41(2) or similar.

In Table 1 there is a category “porous/non-porous”, it should be changed into “silica type”.

In Table 1 there is a category “surface chemistry”. What is the basis of this categorization? What does mean chemical surface SiO2, SiOH? Each silica, regardless of its surface area contains on its surface the silanol groups (SiOH) (less or more, depending on the silica type). It is inappropriate to categorize in this way. Were any studies performed confirming the amount of silanol groups present on the silica surface?

The Authors should describe in more detail the mechanism of silica influence on enzyme activity (the mechanism of action).

Reviewer 2 Report

The aim of this paper was to investigate and describe the mechanism of action of porous silica’s anti-obesity effect using an in vitro digestion. While porous silica is currently used as an anti-obesity therapeutic, the authors of this paper elucidate silica’s mechanism of action upon fat and carbohydrate digestion demonstrating two different modes of action: enzyme activity inhibition and adsorption of macronutrients. This investigation is well designed and structured, supports the authors hypothesis, and eloquently describes the relationship between various porous silica particle physiochemical characteristics and the inhibition of gastrointestinal enzyme activity and adsorption of macronutrients. 

General Comments:

1.     Figure 2 is missing, there are two figures numbered Figure 3.

2.     The n value of 3 is very low, and there are no significant differences described between groups, which would have made for more robust results. Why didn’t authors use n ≥4 so that statistical differences would be described?

3.     The authors use a model of lipase and amylase enzymes at a ratio of 1:1. Is this a physiological ratio of these enzymes in the human gut? Are there any other enzymes that would contribute to fat and carbohydrate digestion. Protein digestion wasn’t addressed in this study. Would the presence of denatured peptides during digestion have any impact upon silica’s enzyme inhibition activity or adsorption?
